# Regulation of Grain Chalkiness and Starch Metabolism by FLO2 Interaction Factor 3, a bHLH Transcription Factor in *Oryza sativa*

**DOI:** 10.3390/ijms241612778

**Published:** 2023-08-14

**Authors:** Xianyu Tang, Weiping Zhong, Kunmei Wang, Xin Gong, Yunong Xia, Jieying Nong, Langtao Xiao, Shitou Xia

**Affiliations:** Hunan Provincial Key Laboratory of Phytohormones and Growth Development, College of Bioscience and Biotechnology, Hunan Agricultural University, Changsha 410128, China; chuxuantingnasha@stu.hunau.edu.cn (X.T.); zweiping@stu.hunau.edu.cn (W.Z.); yuhunan@stu.hunau.edu.cn (K.W.); xingong@stu.hunau.edu.cn (X.G.); yun0623@stu.hunau.edu.cn (Y.X.); njy@stu.hunau.edu.cn (J.N.)

**Keywords:** *Oryza sativa*, *Os*FIF3, chalkiness formation, transcription factor, starch granules, *SUT1*

## Abstract

Chalkiness is a key determinant that directly affects the appearance and cooking quality of rice grains. Previously, *Floury endosperm 2* (*FLO2*) was reported to be involved in the formation of rice chalkiness; however, its regulation mechanism is still unclear. Here, FLO2 interaction factor 3 (*Os*FIF3), a bHLH transcription factor, was identified and analyzed in *Oryza sativa*. A significant increase in chalkiness was observed in *OsFIF3*-overexpressed grains, coupled with a round, hollow filling of starch granules and reduced grain weight. *Os*FIF3 is evolutionarily conserved in monocotyledons, but variable in dicotyledons. Subcellular localization revealed the predominant localization of *Os*FIF3 in the nucleus. The DAP-seq (DNA affinity purification sequencing) results showed that *Os*FIF3 could affect the transcriptional accumulation of *β-amylase 1*, *α-amylase isozyme 2A-like*, *pectinesterase 11*, *β-glucosidase 28 like*, *pectinesterase*, *sucrose transport protein 1* (*SUT1*), and *FLO2* through the binding of the CACGTG motif on their promoters. Moreover, *FLO2* and *SUT1* with abundant *Os*FIF3 binding signals showed significant expression reduction in *OsFIF3* overexpression lines, further confirming *Os*FIF3’s role in starch metabolism regulation and energy material allocation. Taken together, these findings show that the overexpression of *OsFIF3* inhibits the expression of *FLO2* and *SUT1*, thereby increasing grain chalkiness and affecting grain weight.

## 1. Introduction

Rice (*Oryza sativa*) is one of the most important cereal crops worldwide, providing the staple food for nearly half of the global population [1]. During the endosperm filling of rice grains, a large amount of starch synthesis and protein accumulation occurs, and the filling level of the starch component in the endosperm affects both yield and rice quality [2]. When filling is incomplete, too many gaps allow light to pass through the grains and scatter, forming white spots called chalkiness, which is a key determinant of rice yield and nutritional value [3]. Therefore, chalkiness is mainly associated with the filling and formation of starch granules, and understanding the molecular mechanisms regulating rice grain chalkiness and starch metabolism is of vital importance for grain quality and yield improvement [4,5].

Scholars have divided the development of the rice endosperm into four stages: coenocytic, cellularization, storage product accumulation, and maturity [6]. Starch accumulation mainly occurs during the storage product accumulation stage. In the process of endosperm development, the accumulation of starch mainly starts from the central area of the endosperm and gradually spreads outward. After the substantial accumulation of starch and protein, the endosperm begins to dehydrate gradually, and the starch and protein crystallizes during the storage product accumulation and maturity processes, forming a translucent state in the rice grains [7]. 

In 1981, Hikaru Satoh and Hikaru Takeshi obtained some mutants with chalky rice traits through chemical mutagenesis screening; one of these mutants was named *flo2* (*floury endosperm 2*) [8]. It was not until 2010 that the Tokyo University of Science team completed the mapping of this gene, and found that *FLO2* encodes a triangular tetrapeptide repeat (TPR), which to some extent regulates the production of rice chalkiness under condition of high temperature [9]. In addition, the expression of genes related to starch and protein synthesis in the *flo2* mutant is affected to varying degrees. However, the specific molecular mechanism of FLO2 regulation is still elusive.

Many transcription factors have been found to participate in the process of rice grain development and starch metabolism [10]. Among them, basic helix–loop–helix (bHLH) transcription factors form one of the largest transcription factor families and have been demonstrated to play significant roles in various biological processes in plants, including cell division, differentiation, development, and response to environmental stress [11]. Transcription factors that can lead to a chalkiness phenotype, such as NFYB1, RSR1, and bZIP58, are notable. NFYB1 regulates the chalkiness level in rice grains by modulating the expression of *SUT1* (*sucrose transport protein 1*), *SUT3* (*sucrose transport protein 3*), and *SUT4* (*sucrose transport protein 4*), while RSR1 and bZIP58 regulate the chalky phenotype in rice by controlling downstream starch synthase complexes [10]. 

Here, through the screening of FLO2-interacting and chalkiness-related candidates, we identified a new gene and named it *FLO2 interaction factor 3* (*OsFIF3*), which encodes a bHLH transcription factor and plays a vital role in rice grain chalkiness formation and starch metabolism, providing a new insight into the regulatory mechanisms of grain chalkiness.

## 2. Results

### 2.1. Reverse Genetic Analysis Reveals OsFIF3 as a Potential Regulator of Chalkiness Formation

Previously, mutations in *Floury Endosperm 2* (*FLO2*) were shown to enhance chalkiness in rice grains [8], and yeast two-hybrid studies have revealed that some FLO2-interactive proteins exist [9]. By analyzing the published chalkiness-related *FLOs* genes, we here identify a bHLH transcription factor encoding gene, and name it *FLO2 interaction factor 3 (FIF3*) in *Oryza sativa*. The BiFC results showed that FIF3 interacted weakly with FLO2-1 variant translated by the longer transcript, but not with the FLO2-2 variant translated by the shorter transcript (Appendix A). As shown in Figure 1a, the −800~0 bp region of the *OsFIF3* promoter contains an ABA response element, a drought response element, an anaerobic response element, and a meristematic tissue expression element cluster. Among these cis-regulatory elements in the promoters of *OsFIF3* homologous genes across species, the enrichment of the ABA response element is the most significant (Appendix A). According to the Rice Genome Annotation Project, *OsFIF3* is mainly expressed in the pistil and pre-emergence inflorescence, followed by seed-5 DAP, post-emergence inflorescence, and shoots. Additionally, a certain level of expression of *OsFIF3* was also observed in the anther and endosperm-25 DAP (Appendix A) [12].

Phylogenetic analysis and sequence alignment showed that *Os*FIF3 is more conserved in monocotyledons, but variable in dicotyledons (Figure 1a), and shares many conserved sites with Arabidopsis IBH1 (INCREASED LEAF INCLINATION1 BINDING bHLH1) (Figure 1b,c), which is a bHLH transcription factor found to respond to brassinosteroid signaling, with extensive research providing some hints for the function study of *Os*FIF3. However, 24 amino acids were missed at the N-terminal of *Os*FIF3 in the *Oryza sativa Indica* group. Furthermore, there was a significant serine enrichment in the N-terminal 24 amino acids of FIF3 in monocotyledons, which did not exist in dicotyledons (Figure 1c).

### 2.2. Overexpression of OsFIF3 Enhances Chalkiness Degrees in Rice Grains

As *Os*FIF3 is expressed specifically in the reproductive phase of rice, especially in the early grain-filling stage, and its core initiation part of the promoter is enriched with cis-acting elements related to ABA and drought, both of which are related to chalkiness [13,14], we constructed knockout and overexpression constructs of *OsFIF3* to determine its regulatory role in rice chalkiness formation. Consequently, three overexpression lines and three knockout lines of *Os*FIF3 were obtained. As shown in Figure 2a–d, the chalkiness degrees in these overexpressed lines of *OE-FIF3-7*, *OE-FIF3-15*, and *OE-FIF3-16* were 17.49%, 20.02%, and 25.34%, respectively, which were significantly higher than that in WT grains, leading to a belly-white phenotype.

Under scanning electron microscopy (SEM), the sharp-edged appearance of *OE-FIF3-7*, *OE-FIF3-15*, and *OE-FIF3-16* displayed a round filling and a significant number of empty spaces compared with WT starch granules, indicative of incomplete filling, making it difficult for light to pass through (Figure 2i–p). The number of filled starch granules in the cells was also significantly lower than that of the WT grains, with many cells containing only one or two filled starch granules (Figure 2m–p). These unfilled spaces directly contribute to the chalky phenotype.

Then, the expression of *OsFIF3* was quantified by RT-qPCR to determine the relationship between the overexpression level and chalkiness. As shown in Figure 2q, the expression of *OsFIF3* increased by approximately 30, 5, and 3 times, respectively in the flag leaves of *OE-FIF3-7*, *OE-FIF3-15*, and *OE-FIF3-16*. Moreover, at the DAP6 stage of the filled grains, *OsFIF3* expression increased approximately 3~80 times that of the WT grain (Figure 2q). As a result, the chalkiness degrees in grains of *OE-FIF3-7*, *OE-FIF3-15*, and *OE-FIF-16* were 17.49%, 20.02%, and 25.34%, respectively (Figure 2r), showing a significant increase compared with WT grains. Interestingly, a negative correlation between *OsFIF3* expression and grain weight was also revealed (Figure 2s), indicative of an adverse effect of *OsFIF3* overexpression on grain weight. On the contrary, the chalkiness of the knockout mutant grains was not significantly different from the WT grains (Appendix A).

### 2.3. Subcellular Localization of OsFIF3 and Its Interaction DNA Sequence

To determine the subcellular localization of *Os*FIF3, we constructed the eGFP fusion *Os*FIF3 into a binary vector and transiently expressed in the leaves of *Nicotiana benthamiana* via GV3101 infection. By observing DAPI staining co-localization in the abaxial epidermal cells, we confirmed that *Os*FIF3, as a transcript factor, was mainly located in the nucleus (Figure 3).

As *Os*FIF3 shares many conserved sites with *At*IBH1, it is reasonable to speculate that the DNA binding site of *Os*FIF3 might also be CACGTG. Therefore, a yeast one-hybrid experiment was conducted with the pAbai plasmid, which contains the CACGTG sequence as bait, and the pGADT7 plasmid, which contains *Os*FIF3 as prey. As expected, *Os*FIF3 binds to the CACGTG sequence and activates downstream aureobasidin A resistance (Figure 4a–d). Consistently, the EMSA results also showed that *Os*FIF3 bound to the CACGTG probe (Figure 4e). Taken together, these results indicate that *Os*FIF3 is a bHLH transcription factor and binds to the CACGTG motif.

### 2.4. OsFIF3 Regulates Starch Metabolism and Chalkiness Related Gene SUT1

As CACGTG was proven above to be the binding motif of *Os*FIF3, genes that contain the motif in the rice genome were searched and GO/KO analysis was performed (Appendix A). As a result, we found that this motif is enriched in genes involved in certain starch degradation metabolic processes, and starch and sucrose metabolic pathways, including *β-amylase 1* (LOC_Os10g32810), *β-glucosidase 28-like* (LOC_Os08g39870), *pectinesterase* (LOC_Os07g43390), *α-amylase isozyme 2A-like* (LOC_Os06g49970), and *putative pectinesterase 11* (LOC_Os01g44340). Through CUT&TAG using pA-Tn5 with *Os*FIF3-Flag-His, DNA affinity purification sequencing (DAP-seq) in vitro [15,16] was then performed. As shown in Figure 5, large amounts of *Os*FIF3 binding signals on and near the CACGTG motifs were present on these aforementioned genes (Appendix A), indicative of a regulatory role of *Os*FIF3 in starch metabolism.

When these genes were further verified via RT-qPCR, we found that the expression of *β-amylase 1*, *β-glucosidase 28-like*, *pectinesterase*, *α-amylase isozyme 2A-like*, and *putative pectinesterase 11* significantly correlated to *Os*FIF3 binding enrichment. Their expression levels in the flag leaves of *OE-FIF3-7*, *OE-FIF3-15*, and *OE-FIF-16* are significantly higher than that of WT, except for *β-glucosidase 28-like* (Figure 6a–e). However, the expression level of *β-glucosidase 28-like* in *OE-FIF3-16* flag leaves was also significantly higher than that of the WT, and its expression negatively correlated to *OsFIF3* expression. Similar to this, other genes also showed a negative correlation with the expression level of *OsFIF3*, except for *β-amylase 1*, which is positively correlated with *OsFIF3* expression. In addition, DAP-seq results also showed that *Os*FIF3 has two binding sites in the promoter area of *OsFLO2* (Appendix A). When the two transcript variants of *FLO2* were tested, it was found that all of them decreased significantly, although the longer transcript variant *FLO2-1* decreased less. The shorter transcript variant *FLO2-2* reduced about 80% of its expression in these overexpression lines (Figure 6g,h).

Unlike flag leaves, the expression levels of *β-amylase 1*, *β-glucosidase 28-like*, *pectinesterase*, *α-amylase isozyme 2A-like*, and *putative pectinesterase 11* at the DAP6 stage in the overexpression grains showed a significant decrease compared to the WT grains, except for *β-glucosidase 28-like* (Figure 7a–f). Notably, *putative pectinesterase 11* in the grains of *OE-FIF3-7* at the DAP6 stage showed similar expression levels to WT. This might be due to the overexpression of *OsFIF3* inversely suppressing its own expression, or there may be other feedback regulatory mechanisms that exist in the DAP6 stage grains of *OE-FIF3-7*.

Based on the regulatory differences of starch and sugar metabolism-related genes between flag leaves and DAP6-stage grains, it is suggested that *Os*FIF3 might affect the energy substance allocation process and thus lead to chalkiness production. Therefore, we tested the expression level of *SUT1* in flag leaves and DAP6 stage grains. As shown in Figure 6f, there was no significant difference for *SUT1* expression between *OE-FIF3-15*, 16, and WT flag leaves, but the *SUT1* expression in *OE-FIF3-7* was significantly lower than that of the *OE-FIF3-15*, 16, and WT flag leaves. This might also be due to the suppression effects of the extra higher expression of *OsFIF3* in *OE-FIF3-7*. Similarly, *SUT1* expression in the grains of *OE-FIF3-7*, *OE-FIF3-15*, and *OE-FIF-16* was significantly lower than in the WT grains (Figure 7f), indicating that *Os*FIF3 plays a certain regulatory role in the material allocation process of rice plants.

## 3. Discussion

Rice is one of the most crucial staple crops in China, and a chalky phenotype significantly undermines the edible and processing quality of rice grains. The chalky trait in rice is primarily governed by genetics and influenced by environmental factors [7]. In recent years, researchers have unveiled several genetic mechanisms related to chalkiness, including pathways for energy production, as exemplified by FLO10 [17], FLO12 [18], and FLO13 [19]; energy substance allocation, which is represented by NFYB1, GIF1 [10]; starch synthesis, as exemplified by GBSS [20], SS [21], SBE [22], FLO6 [23], RSR1, and bZIP58 [10]; and starch granule accumulation, as exemplified by CHALK5 [24], BIP [25,26], and PDIL1-1 [27]. By analyzing FLO2 interactions, chalkiness-related genes, and the expression pattern of *OsFIF3* in the reproductive phase and early grain-filling stage, we identified *Os*FIF3 as a contributor to the chalky phenotype in rice. *Os*FIF3 is a bHLH transcription factor encoding gene contained an ABA response element, a drought response element in its promoter region of −800 bp, which is consistent with existing reports that ABA and drought can significantly impact the chalkiness level in rice [13,14]. In the early and middle stages of rice grain filling, ABA can promote rice grain filling, while in the later stage, it inhibits it [28,29,30].

FIF3 is relatively conserved in monocots, but more diverse in dicots. However, despite significant differences with the homologs, *Os*FIF3 still contains many conserved sites in dicots. The unique serine-rich region that exists at the N-terminus of monocot FIF3 may suggest its specific function in these species. *Os*FIF3 and *Arabidopsis thaliana* IBH1 [31] are classified as similar transcription factors. Similar to *At*IBH1, we confirmed that *Os*FIF3 has the ability to bind to the CACGTG sequence through yeast one-hybrid and EMSA experiments. The DNA binding signal of *Os*FIF3, however, was not completely coupled to the CACGTG sequence based on the DAP-seq result of *Os*FIF3 using pA-Tn5 in vitro experiments. This may be because *Os*FIF3 not only binds to CACGTG, but also has some affinity for other sequences, similar to MYC2 [32].

The grains of the *OsFIF3* overexpression lines developed a chalky phenotype, and the chalkiness degrees and grain weight tended to decrease with the extra higher expression of *OsFIF3*. The scanning electron microscopy (SEM) results showed that, compared with WT grains, the chalky parts of the starch granules were spherical and loosely stacked; thereby, many pores were created that resulted in the chalky phenotype in *OsFIF3*-overexpressing grains. In the flag leaves of the *OsFIF3* overexpression lines, the expression of *putative pectinesterase 11*, *α-amylase isozyme 2A-like*, *pectinesterase*, *β-glucosidase 28-like*, and *β-amylase 1* were significantly higher than those in the WT, which might cause photosynthetic products to be stored as *β-D-Glucos*e or *α-D-Glucose* instead of starch, making them more likely to participate in transport and biochemical processes. This might explain why the grain weight of *OE-FIF3-16* was significantly higher than that of the WT, as the expression of these related genes in the flag leaves was usually the highest one. However, at the DAP6 stage in grains, except for *β-glucosidase 28-like*, the expression of *putative pectinesterase 11*, *α-amylase isozyme 2A-like*, *pectinesterase*, and *β-amylase 1* showed a decreasing trend compared with WT grains. However, the expression of some starch metabolism-related genes in *OE-FIF3-7* still remained unchanged or were higher compared to the other two lines. This might be the reason why the grain weight of *OE-FIF3-*7 was significantly lower than the WT, while *OE-FIF3-15* exhibited no significant difference with the WT in terms of grain weight.

It has been reported that the absence of *FLO2* led to chalkiness [8,9]. There are two transcript variants of *FLO2*—*FLO2-1* and *FLO2-2*—and the shorter *FLO2-2* transcript variant showed a faster decline than *FLO2-1*. The subcellular localization of these two variants revealed that the faster-declining FLO2-2 was mainly located in the nucleus, while the nuclear localization signal of FLO2-1 was weaker (Appendix A), indicative of the divergent regulation of grain weight and chalkiness by *Os*FIF3 variants, but their specificity needs to be explored and confirmed by further experiments. While the expression of *SUT1* showed a downward expression in grains at the DAP6 stage of all three overexpression lines, a relatively small decline in grains at the DAP6 stage and a significant decline in the flag leaves of *OE-FIF3-*7 was found. This may be due to the regulatory difference between the flag leaves and grains, correlating with the level of *OsFIF3* overexpression. Consistently, NFYB1 was found to regulate chalkiness by controlling sucrose transport proteins like SUT1 [10], and the downregulation of SUT1 can lead to a deficiency of assimilates in grains, thereby causing chalkiness [33]. 

In summary, our data suggested that the overexpression of *OsFIF3* inhibits the expression of *SUT1* and *FLO2*, which are downstream starch and sucrose metabolic genes, thereby increasing the chalkiness level of grains and affecting grain weight.

## 4. Materials and Methods

### 4.1. Plant Materials and Growth Conditions

*OsFIF3* overexpression and CRISPR knockout rice lines were generated from Kitaake tissue culture. The Crispr target design is based on Xie’s methods, and the target sequence was cgggggcttccggcatcgggg, constructed into pYLCRISPR/Cas9-MH with a U3 promoter [34]. All plants were grown under conditions of 12 h light (28 °C)/12 h dark (28 °C), a light intensity of 30,000 Lux, and 70% relative humidity. The samples were planted on soil (nutrient soil: vermiculite: sand = 1:2:3). 

### 4.2. Scanning Electron Microscopy

The rice grains were cross-sectioned, sputter-coated for about 30 s with a Hitachi MC1000 Lon Sputtering Apparatus, and then observed with a Hitachi SU8100 Scanning Electron Microscope.

### 4.3. Vector Construction and Rice Transformation

The cDNA sequence of *OsFIF3* (LOC_Os01g14110) was cloned into the pC1300-eGFP vector driven by the 35S promoter. The plasmid was transferred into GV3101 and then transformed into *Nicotiana benthamiana* leaf for subcellular localization observation. The cDNA sequence of *OsFIF3* was cloned into the pET32a vector driven by the T7 promoter to form the *Os*FIF3-Flag-His fusion protein, and then transferred to BL21. Several BL21 colonies were selected to detect their target protein expression level through SDS-PAGE gel electrophoresis, and the strain with the highest expression level was selected for subsequent experiments.

The cDNA sequence of *FIF3* was cloned into the pC1305-Ubi vector driven by the maize ubiquitin promoter, and the plasmid was transferred into EHA105 and then transformed into Kitaake callus tissue. Positive plants were selected by hygromycin. For CRISPR gene-editing design, the target was determined using Xie’s method and constructed into pYLCRISPR/Cas9-MH with a U3 promoter [34]. This plasmid was transferred to Agrobacterium tumefaciens EHA105 and transformed into Kitaake callus tissue. Positive plants were selected by hygromycin. 

### 4.4. RNA Extraction and RT-qPCR Analysis

The total RNA was extracted from rice tissue using an Eastep™ Super Total RNA Extraction Kit (Promega, Madison, WI, USA), and reverse transcription was performed using the GoScript™ Reverse Transcription System (Promega, Beijing, China). RT-qPCR was performed using 2 × SYBR Green Premix Pro Taq HS Premix (AG11702, Accurate Biotechnology (Hunan) Co., Ltd., Changsha, China)) and a Step-One real-time fluorescence PCR instrument (Applied Biosystems, Bedford, MA, USA). *OsActin1* was used as an internal housekeeping gene. At least three independent biological replicates of each sample were performed.

### 4.5. Subcellular Localization

The *OsFIF3* cDNA was cloned into pC1300-eGFP to construct a fusion protein with eGFP. This plasmid was transformed into Agrobacterium tumefaciens GV3101 and transiently expressed in tobacco leaves. The eGFP fluorescence signal was detected using a Zeiss LSM710.

### 4.6. EMSA

The biotin-labeled probe was synthesized by a biotechnology company. The *OsFIF3* cDNA sequence was cloned into the pET32a vector to generate the FIF3-Flag-HIS fusion protein, which was then transformed into BL21 *E. coli* and expressed overnight at 20 °C with 1 mM IPTG induction. The cells were treated with a KS-650ZDN ultrasonicator and the target protein was purified using HIS magnetic beads purification. By adding 2-fold and 5-fold unlabeled probes to compete for the binding of biotin labeled probes, the *Os*FIF3 binding signal of biotin labeled probes will weaken accordingly as the concentration of unlabeled probes increases.

### 4.7. DAP-Seq

The *OsFIF3* cDNA sequence was cloned into the pET32a vector to construct the FIF3-Flag-His fusion vector, which was then transferred to BL21. The protein was induced to express overnight at 20 °C with 1 mM IPTG. The cells were treated with a KS-650ZDN ultrasonicator and the target protein was purified with His magnetic beads. After purification, the *Os*FIF3 protein was incubated with 10 μg Kitaake genomic DNA at room temperature for 1 h, and then incubated with the Flag monoclonal antibody at room temperature for 1 h. After that, the *Os*FIF3 fusion protein was incubated with His magnetic beads for 30 min and washed three times with His wash buffer for 10 min each. A secondary antibody was added and incubated at room temperature for 1 h. The protein was then washed three times, each for 10 min, with His wash buffer. The *Os*FIF3 fusion protein was then eluted with His elution buffer, and the DNA fragments adsorbed by the target protein were amplified using the assembled pA-Tn5 library end Index primers. After amplification, three parallel samples in equal amounts of a mixture were sent to Sangon Biotech (Shanghai, China) for high-throughput sequencing.

The original sequencing data were subjected to statistics and quality assessment through Fastp, including adapter removing, quality, length, and low-complexity filtering, per read cutting by quality below 20Q, base correction for PE data, as well as pollution assessment, which was conducted by randomly selecting 10,000 reads from each sequencing sample, comparing them to the NT library through blastn, and then merging the results to obtain the abundance of the species and assessing whether there was contamination from other species (Appendix A).

The clean sequencing data were then compared to the Kitaake genome using bwa (0.7.15) software [35]. The macs2 (2.2.9.1) software was applied to peak calling and SAMtools (1.15.1) software was used to convert them into bed files for analysis on IGV (2.3.79) software [36,37]. CentriMo analysis was performed on values with fold enrichment > 13 [38].

### 4.8. Bioinformatics Analysis

A total of 2000 bp sequences from the FIF3 promoter were submitted to the PlantCare website (http://bioinformatics.psb.ugent.be/webtools/plantcare/html, accessed on 14 April 2023) for Cis-regulatory element prediction [39]. Using CARMO software (http://bioinfo.sibs.ac.cn/carmo/Gene_Annotation.php, accessed on 11 May 2023), the GO and KO analysis of genes containing the CACGTG sequence within the 2000 bp promoter in the rice genome was conducted. The amino acid sequences of *Os*FIF3 homologs were aligned using Cluster X [40], and MEGA7 (7.0.25) software was used for phylogenetic analysis [41]. The Kitaake genome sequence was downloaded from the phytozome site (https://phytozome-next.jgi.doe.gov/info/OsativaKitaake_v3_1, accessed on 29 October 2021). Blastp was used for homolog screening (non-redundant RefSeq proteins).

## Figures and Tables

**Figure 1 ijms-24-12778-f001:**
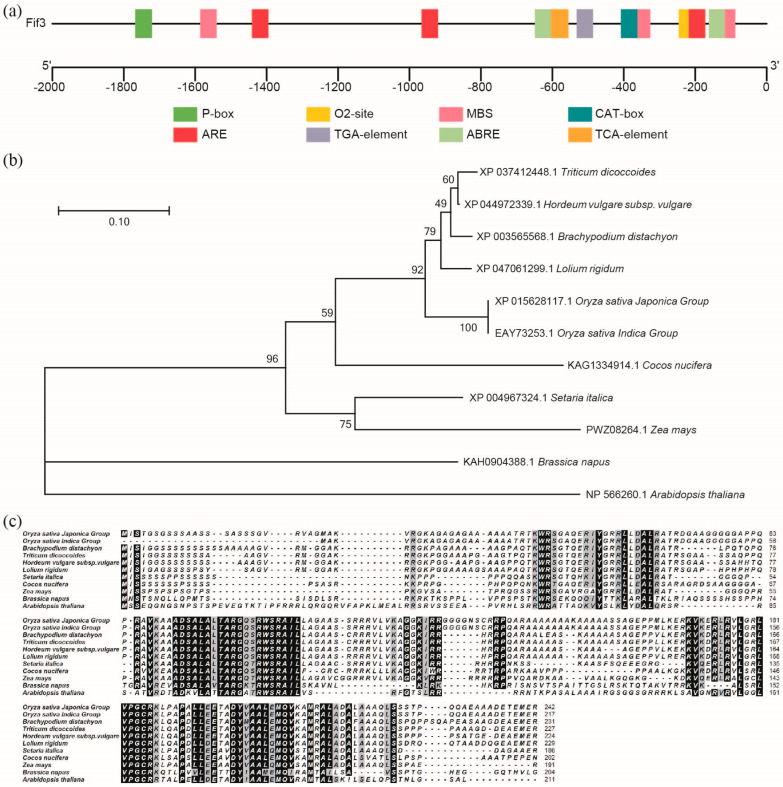
Gene structure and phylogenetic analysis of *Os*FIF3. (**a**) Cis-regulatory element analysis of the *OsFIF3* promoter. P-box, the gibberellin-responsive element; O2-site, a cis-acting regulatory element involved in zein metabolism regulation; MBS, the MYB binding site involved in drought-inducibility; CAT-box, a cis-acting regulatory element related to meristem expression; ARE, a cis-acting regulatory element essential for anaerobic induction; TGA-element, an auxin-responsive element; ABRE, a cis-acting element involved in the abscisic acid responsiveness; TCA-element, a cis-acting element involved in salicylic acid responsiveness. These cis-regulatory elements are enriched within −800 bp upstream of the start codon of *OsFIF3*. (**b**) Phylogenetic analysis (the numbers of percentages indicate the bootstrap value) and (**c**) sequence alignment of *Os*FIF3. *Os*FIF3 homologs from *Oryza sativa Indica*, *Brachypodium distachyon*, *Triticum dicoccoides*, *Hordeum vulgare* subsp. *vulgare*, *Lolium rigidum*, *Setaria italica*, *Cocos nucifera*, *Zea mays*, *Brassica napus*, and *Arabidopsis thaliana* were blasted and analyzed.

**Figure 2 ijms-24-12778-f002:**
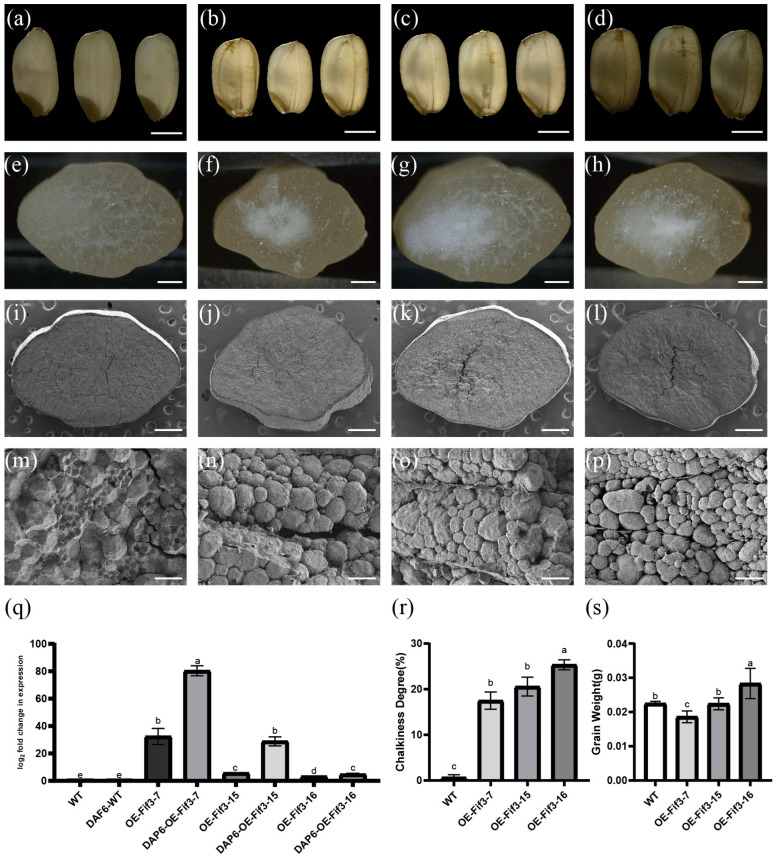
Grain phenotype of *OsFIF3* overexpression lines. Grains of (**a**) WT, (**b**) *OE-FIF3-7*, (**c**) *OE-FIF3-15*, and (**d**) *OE-FIF3-16* overexpression lines under backlight conditions are displayed. The chalky area, which obstructs light transmission, is presented as a dark shadow. Bar = 2 mm. Cross-sectioned grains of (**e**) WT, (**f**) *OE-FIF3-7*, (**g**) *OE-FIF3-15*, and (**h**) *OE-FIF3-16* overexpression lines: the chalky part is shown as white spots. Bar = 500 μm. Cross-section of (**i**,**m**) WT, (**j**,**n**) *OE-FIF3-7*, (**k**,**o**) *OE-FIF3-15*, and (**l**,**p**) *OE-FIF3-16* overexpression grains under scanning electron microscopy (SEM). The chalky areas of *OE-FIF3-7* (**n**), *OE-FIF3-15* (**o**), and *OE-FIF3-16* (**p**) display a large number of non-tightly packed round starch granules, thus causing a large number of gaps without filled starch. (**i**–**l**) Bar = 500 μm and (**m**–**p**) bar = 15 μm. (**q**) *OsFIF3* expression levels in WT, *OE-FIF3-7*, *OE-FIF3-15*, and *OE-FIF3-16* overexpression lines in flag leaves and endosperm during DAP6 (6 days after pollination) stage. Different lowercase letters indicate significant difference at the 0.05 level. (**r**) Chalkiness degree levels in WT, *OE-FIF3-7*, *OE-FIF3-15*, and *OE-FIF3-16* overexpression lines. Different lowercase letters indicate significant difference at the 0.05 level. (**s**) Grain weight of WT, *OE-FIF3-7*, *OE-FIF3-15*, and *OE-FIF3-16* overexpression lines. Different lowercase letters indicate significant difference at the 0.05 level.

**Figure 3 ijms-24-12778-f003:**
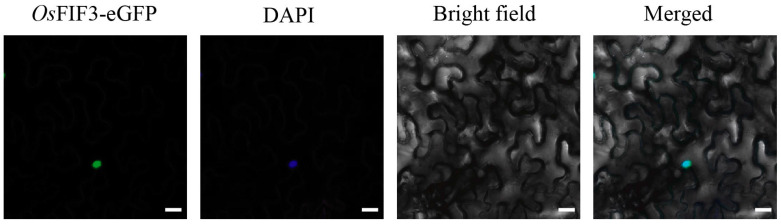
Subcellular localization of *Os*FIF3. The signal of *Os*FIF3-eGFP (green) overlaps with the nuclear DAPI dye (blue). Bar = 10 μm.

**Figure 4 ijms-24-12778-f004:**
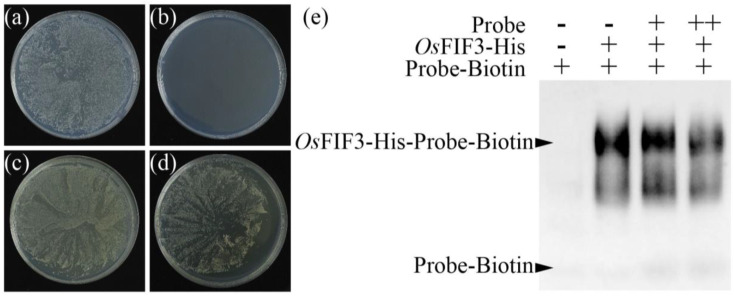
Yeast one-hybrid and EMSA analysis of *Os*FIF3. Left, yeast one-hybrid analysis of *Os*FIF3. (**a**) Y1HGold with CACGTG motif-Abai grown on Ura-deficient medium. (**b**) Y1HGold with CACGTG motif-Abai cannot grow on Ura-deficient medium supplemented with aureobasidin. (**c**) Y1HGold with *Os*FIF3-AD grown on Leu-deficient medium. (**d**) Y1HGold with both *Os*FIF3-AD and CACGTG motif-Abai grown on Leu-deficient medium supplemented with aureobasidin. Right, EMSA analysis of *Os*FIF3. As the concentration of unlabeled biotin probe increases, the signal binding to the biotin probe weakens (**e**).

**Figure 5 ijms-24-12778-f005:**
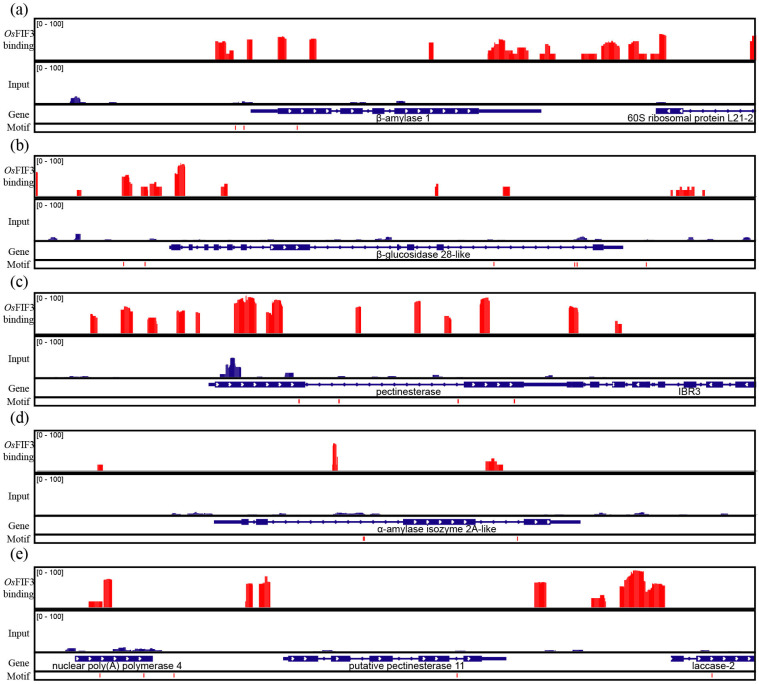
CUT&TAG and DAP-seq of *Os*FIF3. Enrichment of *Os*FIF3 binding sites at promoter and sequence area of the following genes: (**a**) *β-amylase 1*, (**b**) *β-glucosidase 28-like*, (**c**) *pectinesterase*, (**d**) *α-amylase isozyme 2A-like*, and (**e**) *putative pectinesterase 11*. Red peaks indicating reads of *Os*FIF3 binding abundance.

**Figure 6 ijms-24-12778-f006:**
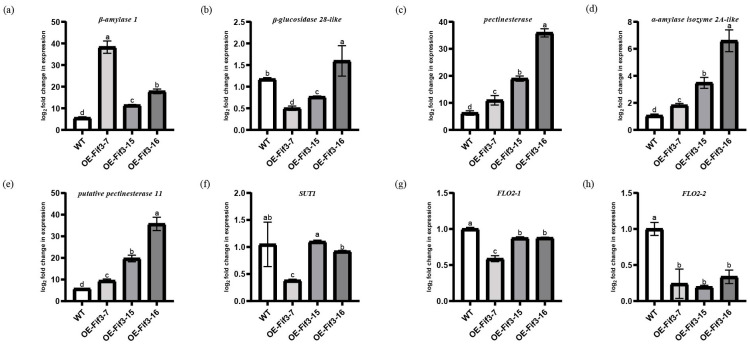
Expression levels of starch metabolism and chalkiness-related genes in leaves of *OsFIF3* overexpression lines. (**a**) Expression level of *β-amylase 1*, (**b**) *β-glucosidase 28-like*, (**c**) *pectinesterase*, (**d**) *α-amylase isozyme 2A-like*, (**e**) *putative pectinesterase 11*, (**f**) *SUT1*, (**g**) *FLO2-1*, and (**h**) *FLO2-2* in flag leaves. The different lowercase letters indicate significant difference at the 0.05 level.

**Figure 7 ijms-24-12778-f007:**
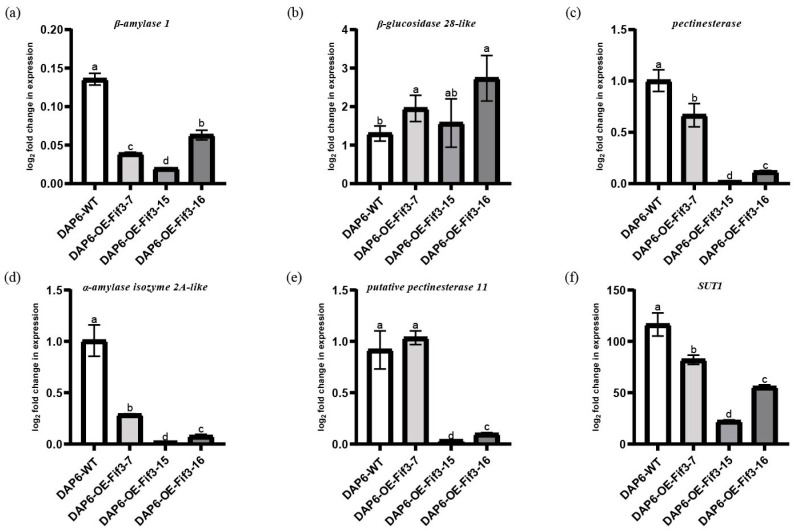
Expression levels of starch metabolism and chalkiness-related genes in grains of *OsFIF3* overexpression lines. Expression level of (**a**) *β-amylase 1*, (**b**) *β-glucosidase 28-like*, (**c**) *pectinesterase*, (**d**) *α-amylase isozyme 2A-like*, (**e**) *putative pectinesterase 11*, and (**f**) *SUT1* in DAP6 grains. The different lowercase letters indicate significant difference at the 0.05 level.

## Data Availability

Data files are available upon request. Clean DAP-seq data were uploaded to SRA PRJNA998089.

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
