# Peer review of "Regulation of Grain Chalkiness and Starch Metabolism by FLO2 Interaction Factor 3, a bHLH Transcription Factor in *Oryza sativa"

_ijms, 2023, doi:10.3390/ijms241612778_

Round 1
Reviewer 1 Report
This manuscript by Tang et al. tried to identify a transcription factor that regulates FLO2 gene affecting chalkiness in rice grains. The authors investigated one transcription factor among reported genes interacting with FLO2 from mutant rice lines. With several trials of investigation, authors named this gene OsFIF3 and showed phenotypic changes in the chalkiness of rice from OsFIF3 mutant lines. From this study, novel DAP-seq data was generated. Phylogenetic tree analysis was performed with protein sequences from different species to observe the sequence similarity across species. Cis-regulatory elements were identified on upstream of OsFIF3. Also, several target genes of OsFIF3 were introduced and their expression patterns were detected in OsFIF3 overexpression lines.
Since rice is one of the major staple food, the improvement of the chalkiness of rice grain can impact the quality of rice and its economic value. To find the regulator of FLO2, OsFIF3 was focused on and investigated in this study. Identifying the functions of a gene can be novel and valuable. However, there are logical flaws in this manuscript and many key pieces of information are hidden or neglected to support their findings and statements. Thus, I would recommend reconsidering this manuscript after major revision.
Please see the attached file for the major concerns and other comments.

Author Response
We genuinely thank you for your constructive and helpful suggestions and comments, which help us to improve the quality and thoroughness of our study. The manuscript texts have since been revised substantially. Some minor mistakes and typos were also corrected during revision. Our response in red font can be found beneath each original reviewer’s comment below.

Reviewer 2 Report
The manuscript describes the function of bHLH transcription factor OsFIF3, which interacts with FLO2 regulating rice grain chalkiness. Overexpression of OsFIF3 enhances the chalkiness of rice grain. Yeast one-hybrid assay and EMSA shows OsFIF3 binds to the CACGTG motif. DNA-affinity purification sequencing reveals the target genes of OsFIF3. These results show the repressive role of OsFIF3 in grain chalkiness via regulating FLO2 and SUT1. However, several issues should be addressed in the manuscript before acceptance, as listed below.
Major points
1) The interaction between FLO2 and OsFIF3 has been shown in previous yeast two-hybrid assays, but no other experimental evidence is presented in this manuscript. Multiple methods should usually analyze protein-protein interactions; additional experiments such as BiFC, pull-down assay, and co-immunoprecipitation are desirable.
2) In Figure 4, the sequence of the DNA incorporated into the Y1H bait vector and the EMSA probe is not available. Are these short sequences containing only the CACGTG motif?
Both experiments in Figure 4 should be performed with mutations introduced in the CACGTG motif, which could demonstrate the binding specificity of OsFIF3.
3) In the DAP-seq in Figure 5, any statistical analysis has performed to show the OsFIF3 binding region? It is necessary to determine whether the signal is statistically significant enrichment. If statistical analysis is difficult, quantitative PCR should be used to confirm OsFIF3 binding.
Minor point
4) Line 18:Please spell out "DAP-seq".
Author Response

(The authors gave the same response as above.)

Reviewer 3 Report
The paper ”Regulation of Grain Chalkiness and Starch Metabolism by FLO2interaction factor 3, bHLH transcription Fatcor in Oryza sativa” by Xianyu Tang et al. identified the OsFIF3, an interactor of FLO2 as one of genes related to the grain chalkiness. the authors produced both overexpression and knockout rice plants of OsFIF3 and found that overexpression of OsFIF3 caused the grain chalkiness. moreover, the authors analyze the downstream target genes and found that OsFIF3 bound CACGTG motif in the promoters of b-amylase 1, a-amylase isozyme 2A like, and so on and regulated the expression of these genes. in general, the results are important in this field and interesting. However, there are several consernes.
1. L71-72: the authors stated that they found OsFIF3 in the paper (reference No6). Is it AK070651 shown in the reference paper 6? Because this part is hard to understand. In addition, the authors wrote “ there are some FLO2 interactive gene existed(6) (in L72). Aren’t they interacting proteins with FLO2 (because they were identified by Y2H)? The sentence is very confusing.
2. Figure 1. I cannot understand why the authors show the binding site of transcription factors in the promoter of OsFIF3. I think it is better to place the last part (C). phylogenetic tree should be first (A).
Is XP015628117 OsFIF3? The authors should note which protein is OsFIF3 and AtIBH1 in the phylogenetic tree. In addition, it is better to write the RAP number or something somewhere.
3. L96: the authors mention Arabidopsis IBH1 without any explanation. Add explanation about the gene.
4. figure 2. The categories on x-axis are hard to see. and because the phenotype of WT were shown on the far left, it is better to put WT on the far left in the graphs.
5. Figure 4 (e): a whole blot should be shown in the case of EMSA.
minor
1. there are several sentences with unfamiliar expression which I haven’t seen in papers in this field.
For example, L38: Scholars have divided the development
L46: Hiraku Satoh obtained,,,,
2. L48: I think Nippon institute of technology should be Tokyo university of science. Please check it.
3.the authors mention NFYB1, RSR1, SUT1, SUT3,,, in the introduction section without no explanation. We cannot understand what they are. Please explain it.
Author Response

(The authors gave the same response as above.)

Round 2
Reviewer 1 Report
Thank you for your responses. Many concerns and questions have been addressed in the responses. However, I have new concerns, suggestions, and questions based on the updated information and responses from the authors.
Moreover, some responses and the updated manuscript still have a typo or flaws. Also, the authors only explained the information in this document to a reviewer, but haven’t included them in the manuscript. For these cases, I added new comments and opinions in Feedback.
Please find details in the attached document.

Author Response
Thanks for your comments and suggestions, we have revised the manuscript accordingly, please see the details in the attached file.

Reviewer 2 Report
The revised manuscript has adequately addressed the issues pointed out in the previous review.
Author Response
Thanks again for your comments and suggestions.
Reviewer 3 Report
Fig. 4 please remove the old version of (e) (upper one)
The authors have addressed all my concerns.
Author Response
Thanks for your feedback, old version had been deleted.